# Immunization with *Brucella abortus* S19Δ*per* Conferred Protection in Water Buffaloes against Virulent Challenge with *B. abortus* Strain S544

**DOI:** 10.3390/vaccines9121423

**Published:** 2021-12-02

**Authors:** Pallab Chaudhuri, Mani Saminathan, Syed Atif Ali, Gurpreet Kaur, Shiv Varan Singh, Jonathan Lalsiamthara, Tapas K. Goswami, Ashwini K. Singh, Sandeep K. Singh, Praveen Malik, Raj K. Singh

**Affiliations:** 1Division of Bacteriology and Mycology, ICAR-Indian Veterinary Research Institute, Izatnagar, Bareilly, Uttar Pradesh 243122, India; atifali@gate.sinica.edu.tw (S.A.A.); 22gurpreet1990@gmail.com (G.K.); shivvaransingh1@gmail.com (S.V.S.); 2Division of Pathology, ICAR-Indian Veterinary Research Institute, Izatnagar, Bareilly, Uttar Pradesh 243122, India; m.saminathan@icar.gov.in; 3Department of Molecular Microbiology and Immunology, Oregon Health and Science University, Portland, OR 97239, USA; lalsiamt@ohsu.edu; 4Immunology Section, ICAR-Indian Veterinary Research Institute, Izatnagar, Bareilly, Uttar Pradesh 243122, India; tapas.goswami@icar.gov.in; 5Chaudhary Charan Singh National Institute of Animal Health, Baghpat, Uttar Pradesh 250609, India; ashwini.kumar.niah@gov.in (A.K.S.); sandeep.niah@gov.in (S.K.S.); praveen.malik.niah@gov.in (P.M.); 6Division of Biotechnology, ICAR-Indian Veterinary Research Institute, Izatnagar, Bareilly, Uttar Pradesh 243122, India; rks_virology@rediffmail.com

**Keywords:** brucellosis, live attenuated vaccine, S19Δ*per*, buffalo, DIVA, protective efficacy

## Abstract

Vaccination of cattle and buffaloes with *Brucella abortus* strain 19 has been the mainstay for control of bovine brucellosis. However, vaccination with S19 suffers major drawbacks in terms of its safety and interference with serodiagnosis of clinical infection. *Brucella abortus* S19∆*per*, a perosamine synthetase *wbk*B gene deletion mutant, overcomes the drawbacks of the S19 vaccine strain. The present study aimed to evaluate the potential of *Brucella abortus* S19Δ*per* vaccine candidate in the natural host, buffaloes. Safety of S19∆*per*, for animals use, was assessed in guinea pigs. Protective efficacy of vaccine was assessed in buffaloes by immunizing with normal dose (4 × 10^10^ colony forming units (CFU)/animal) and reduced dose (2 × 10^9^ CFU/animal) of S19Δ*per* and challenged with virulent strain of *B. abortus* S544 on 300 days post immunization. Bacterial persistency of S19∆*per* was assessed in buffalo calves after 42 days of inoculation. Different serological, biochemical and pathological studies were performed to evaluate the S19∆*per* vaccine. The S19Δ*per* immunized animals showed significantly low levels of anti-lipopolysaccharides (LPS) antibodies. All the immunized animals were protected against challenge infection with *B. abortus* S544. Sera from the majority of S19Δ*per* immunized buffalo calves showed moderate to weak agglutination to RBPT antigen and thereby, could apparently be differentiated from S19 vaccinated and clinically-infected animals. The S19Δ*per* was more sensitive to buffalo serum complement mediated lysis than its parent strain, S19. Animals culled at 6-weeks-post vaccination showed no gross lesions in organs and there was comparatively lower burden of infection in the lymph nodes of S19Δ*per* immunized animals. With attributes of higher safety, strong protective efficacy and potential of differentiating infected from vaccinated animals (DIVA), S19Δ*per* would be a prospective alternate to conventional S19 vaccines for control of bovine brucellosis as proven in buffaloes.

## 1. Introduction

Brucellosis is a major zoonotic disease caused by the *Brucella* species, which remains an uncontrolled problem in different developing countries, including India. The major clinical manifestations of brucellosis in animals are infertility, repeat breeding, retention of placenta, abortion and stillbirth [1,2]. Bovine brucellosis is endemic in India [3,4] causing severe economic losses to the tune of USD 3.4 billion every year to the livestock industry [5]. Humans get infections from infected animals and contaminated animal products causing undulant fever, orchitis, chills, fatigue and joint and muscle pain [6,7].

Animal vaccination, serological examination and culling of infected animals play pivotal roles in successful control and eradication of brucellosis [8]. Following the strategy of test-slaughter and compensation, a number of countries have effectively eradicated animal brucellosis and achieved the status of being brucellosis-free regions [9,10,11,12]. However, in developing countries, the elimination of infected animals is not practicable due to economic burden and cultural belief restricting cow slaughter and thus, leading to high prevalence of brucellosis in these countries. Vaccination of animals, thereby, continues to be the keystone of controlling animal brucellosis in these areas. Since, there is no human vaccine available for brucellosis, animal vaccination is crucial to control and reduce the incidences of brucellosis in humans. Vaccination of domestic animals is typically performed by the administration of live attenuated *Brucella* strains, such as *B. abortus* S19 (S19) for bovines and *B. melitensis* Rev.1 (Rev.1) for small ruminants [13]. Although S19 and Rev.1 vaccines have been successfully used worldwide, major drawbacks of these strains are presence of residual virulence and interference in serodiagnosis of the disease and inability of differentiating infected from vaccinated animals (DIVA).

In the previous study, we reported the development of a perosamine synthetase *wbk*B gene (Locus Tag-BAbS19_105070) deletion mutant of *Brucella abortus* strain S19, named as S19Δ*per*. The S19Δ*per* with altered lipopolysaccharide (LPS) conferred solid protection to immunized mice against virulent challenge [14]. The whole genome sequence of S19Δ*per* confirmed the complete deletion of perosamine synthetase *wbk*B gene with replacement of kanamycin cassette [15].

Mouse remains the most preferred animal model for understanding immune response to *Brucella* vaccines [16,17]. However, results of mice experiments may not be extrapolated to determine the immune responses in natural hosts, cattle and buffaloes. India has the largest buffalo population (109.85 million) in the world, contributing the major share in milk production in the country. With endemicity of bovine brucellosis in India [3,4], it is warranted to assess the efficacy of S19Δ*per* in the natural host. This study aimed to evaluate the protective immune responses and DIVA status of S19Δ*per* strain in buffaloes (*Bubalus bubalis*). The safety of S19Δ*per* strain has also been evaluated in in vitro and in vivo studies.

## 2. Materials and Methods

### 2.1. Biosafety and Compliance with Animal Ethics

*Brucella* organisms were handled and processed in accordance with the guidelines provided by the Institute Biosafety Committee (IBSC), ICAR-Indian Veterinary Research Institute (ICAR-IVRI), Izatnagar, India. Animal experiments on guinea pigs and buffaloes were performed with the approval and following of the guidelines of the Institute Animal Ethics Committee (IAEC), ICAR-IVRI and the Committee for the Purpose of Control and Supervision of Experiments on Animals (CPCSEA), Ministry of Fisheries, Animal Husbandry and Dairying, New Delhi, India. A challenge study on buffalo calves was carried out at Chaudhary Charan Singh National Institute of Animal Health (CCSNIAH), Baghpat, India, with necessary permissions from IBSC and IAEC of CCSNIAH and CPCSEA.

### 2.2. Bacterial Strains, Media and Growth Conditions

Reference vaccine strain, S19, and challenge strain, *B. abortus* 544 (S544), used in this study were obtained from Division of Biological Standardization and *Brucella* Reference Laboratory, ICAR-IVRI, respectively. *Brucella abortus* S19∆*per* used in this study was developed in our previous study by deletion of perosamine synthetase *wbk*B gene from the S19 strain [14,15]. S19∆*per* possesses an altered LPS. Organisms were grown routinely in BBL-*Brucella* agar (BBA) and tryptose phosphate broth (TPB). The S19 and S19∆*per* strains were incubated in aerobic conditions, while S544 was grown under 5% CO_2_ atmosphere at 37 °C. Erythritol (2 mg/mL) was supplemented on BBA to differentiate S19∆*per* and S19 from S544. S19∆*per* and S19 are sensitive to erythritol.

### 2.3. Safety Assay of S19∆per in Guinea Pigs

Before initiation of vaccine trial in buffaloes, safety of S19∆*per* was proven in experimental guinea pig model. Six adult female guinea pigs were injected intramuscularly with 5 × 10^9^ colony forming units (CFU) S19Δ*per*, a recommended dose mentioned in Indian Pharmacopoeia (IP). Animals were observed for 10 d as per the IP guidelines for development of any clinical signs and mortality. Further, the animals were sacrificed and gross pathological changes in different organs were recorded. Different organs viz. axillary and inguinal LNs, spleen, thymus, liver, kidneys, uterus, ovaries, lungs, heart and brain were collected aseptically for bacterial isolation and preserved in 10% NBF for histopathology.

### 2.4. Experimental Vaccine Preparation, Immunization and Challenge Infection in Buffaloes

*Brucella abortus* S19 and S19Δ*per* were cultivated on *Brucella* agar medium for 72 h and the cells were harvested in phosphate buffer saline (PBS). A ten-fold serial dilution of harvested cells was made in PBS and bacterial count was determined. Cell concentration in the S19 and S19Δ*per* vaccine formulations were adjusted to 4 × 10^10^ colony forming units (CFU)/dose. A reduced dose (1–2 × 10^9^ CFU) of S19Δ*per* was also prepared for immunization. A total of 24 female Murrah buffalo calves, 6–8 months age were used in the present study. Among these animals, a group of 12 buffaloes were used for immunization and challenge study while another group of 12 buffaloes were used for vaccine bacterial persistence assay. Prior to the experiments, seronegativity was confirmed on these animals using rose bengal plate agglutination test (RBPT) and standard tube agglutination test (STAT). All the animals were given a deworming dose of Fentas Plus, Intas Pharmaceuticals Ltd., Ahmedabad, India (contains fenbendazole and praziquantel), before the commencement of the experiments. Animals were fed a balanced diet of concentrate, mineral salt mixture and green fodder. Access to veterinary care and animal wellbeing were regularly monitored. The scheme and experimental layout are depicted in Figure 1.

Twelve buffalo calves were randomly distributed into 4 different groups. Animals were immunized via subcutaneous (S/C) route as follows: Group I, S19 (4 × 10^10^ CFU); Group II, S19Δ*per* normal dose (4 × 10^10^ CFU); Group III, S19Δ*per* reduced dose (2 × 10^9^ CFU); and Group IV, PBS control. Serum samples were collected at specified time intervals. Challenge infection was given 300 d post immunization. All the animals were given challenge infection with a virulent strain of *B. abortus* S544 through orbital route. A volume of 50 µL of culture was instilled in the lower conjunctival sac of both eyes of the buffaloes. Challenge dose contained a total of 2 × 10^7^ live S544 organisms. To measure load of infection of S544 strain, animals were sacrificed after 30 d of challenge infection. Spleen, liver and lymph nodes (LNs) were aseptically collected. Tissue samples were macerated in a mortar and pestle and diluted in PBS. The triturate was placed, as small droplets using micropipette on BBA plates supplemented with erythritol and antibiotics (vancomycin, nystatin and polymyxin B) to prevent the growth of contaminating organisms. Plates were incubated at 37 °C for 5 d in a CO_2_ incubator for visible growth of *B. abortus* S544 colonies and accordingly the bacterial load was calculated for each one gram of tissue.

### 2.5. Indirect ELISA for the Detection of Anti-Brucella Antibodies

Sonicated antigen was prepared from *B. abortus* S99 strain and used as coating antigen. A 96-well polystyrene microplate (Nunc, Waltham, MA, USA) was coated with 100 μL of antigen (1 µg/mL) in 0.05 M carbonate buffer, pH 9.6 for 18–24 h at 4 °C. All the wells were rinsed five times with the PBST solution (0.1 mM disodium hydrogen orthophosphate and 0.05% Tween 20). The control and test sera were diluted 1:200 in the diluting buffer (PBS, 0.05% Tween 20, pH 7.2) and 100 μL diluted serum samples were applied to respective wells in duplicate. The plates were then incubated for 30 min at room temperature (RT) on a rotary shaker. A volume of 100 μL of rabbit anti-bovine horseradish peroxidase (HRPO) conjugate (Sigma-Aldrich, St. Louis, MO, USA) was added and incubated for 1 h at RT. Finally, 100 μL of chromogenic substrate, OPD (10 mg dissolved in 25 mL of 0.05 M phosphate-citrate buffer, pH 5.5 with 10 μL of 30% H_2_O_2_) was added to each well. The plates were placed on an orbital shaker for 15 min prior to reading at 450 nm in a microplate reader. The degree of the color that develops (optical density measured at 450 nm) is directly proportional to the amount of antibody specific to *B. abortus* present in the sera of animals.

### 2.6. Serum Biochemical Profiles

Biochemical parameters, namely, aspartate amino transferase (AST) or serum glutamic oxaloacetic transaminase (SGOT), alanine amino transferase (ALT) or serum glutamic-pyruvic transaminase (SGPT), alkaline phosphatase (AP), total protein, albumin, direct bilirubin, total bilirubin, creatine phosphokinase (CPK) and creatinine were estimated in serum samples of vaccinated animals using kits of Coral Clinical Systems (Tulip Diagnostics Pvt. Ltd., Goa, India) as per manufacturer’s protocol. The OD values were measured in a Multiskan Go spectrophotometer (Thermo Scientific, Carlsbad, CA, USA).

### 2.7. Complement Sensitivity Assay

Sensitivity of *B. abortus* S19 and S19Δ*per* to serum complement was tested by incubating the organism in serum collected from 12 buffalo calves. Freshly collected buffalo serum was first tested for negativity to RBPT antigen. Live *Brucella* cells (1 × 10^5^ CFU) were incubated with 100 µL of serum complement (1:4 dilution) for 1 h at 37 °C. Similarly, cells were also incubated with decomplemented buffalo serum and PBS as controls. After incubation, each of the test lots were spread on BBA plates and incubated at 37 °C for 72 h.

### 2.8. RBPT and DIVA Assessment

Blood samples (*n* = 24) were collected from all the groups of vaccinated buffaloes at different time points. The RBPT was performed using a volume of 30 µL of test serum sample and 30 µL of RBPT test antigen. Formation of agglutination is the indicator of positive reaction. Degree of agglutinations were qualitatively termed as strong (+++), moderate (++), weak (+) and no reactivity (−). For DIVA assessment, RBPT was done on sera collected after 6 weeks of immunization. Sera samples from clinically positive animals were used for comparison.

### 2.9. Bacterial Persistency

Twelve buffalo calves were randomly assigned into 3 groups with 4 animals in each group. Animals were inoculated either with S19 or S19Δ*per* in 2 mL volume through S/C route in the brisket region of the neck. Group I was administered with S19 (4 × 10^10^ CFU), Group II with S19Δ*per* normal dose (4 × 10^10^ CFU) and group III with a reduced dose of S19Δ*per* (1 × 10^9^ CFU). To measure the persistence of *Brucella* strains *in vivo*, animals were sacrificed after 42 d of inoculation. The spleen, liver and lymph nodes (LNs) were aseptically collected. Persistence of vaccinating organisms in the collected organs was determined. Tissue samples were macerated in a mortar and pestle and diluted in PBS. The triturate was placed, as small droplets, using a micropipette onto BBA plates supplemented with antibiotics (vancomycin, nystatin and polymyxin B) to prevent the growth of contaminating organisms. The plates were incubated at 37 °C for 5 d for visible growth of *Brucella* colonies and thereby to measure the level of persistence of S19 and S19Δ*per* in different organs.

### 2.10. Necropsy Examination

The animals were euthanized by intravenous administration of overdose of thiopentone sodium. Systematic necropsy examination was conducted after euthanasia and gross lesions in different organs were recorded. During necropsy, different lymph nodes (LNs) viz. prescapular, parotid, cervical, mandibular, retropharyngeal, supramammary, prefemoral (precrural), hepatic, mesenteric and mediastinal LNs, and spleen (parameters such as measurement of spleen using scale and appearance of borders were used to define the enlargement of spleen), tonsil, liver, kidneys, uterus, ovaries, lungs, heart and brain were collected aseptically from buffaloes for bacterial isolation and preserved in 10% neutral buffered formalin (NBF) for histopathological examination.

### 2.11. Histopathological Studies

The formalin-fixed tissue samples were cut into pieces of 2–3 mm thickness and washed thoroughly with water, dehydrated in ascending grades of alcohol and cleared in xylene. The dehydrated tissues were embedded in paraffin blocks. Sections of 4 µm thickness were cut and stained with hematoxylin and eosin (H&E) stain. Under low power objective (4×), the lymphoid follicles were counted in 10 representative fields in spleen and LNs.

### 2.12. Statistical Analysis

Statistical analyses were used wherever applicable. Two-way analysis of variance (ANOVA) with Tukey’s multiple comparison posthoc test and unpaired t-test were used to determine statistically significant differences. The means of the different groups at specific time intervals were expressed as mean ± standard deviation (SD). The differences were considered as significant with *p*-values of ≤0.05 (*) or ≤0.0001 (****).

## 3. Results

### 3.1. Gross and Histopathological Lesions of Vaccine Safety Assay in Guinea Pigs

Following the guidelines of IP for safety study, a group of six guinea pigs were injected with S19Δ*per*. None of the animals showed any clinical signs or mortality. The seed culture of S19∆*per* passes the safety test. The spleen and LNs of vaccinated guinea pigs showed no obvious gross lesions, especially enlargement. No visible gross lesions were observed in other internal organs like the thymus, liver, lungs, heart, kidneys, ovaries, uterus and brain. The spleen, LNs, thymus, liver, lungs, heart, kidneys, ovaries, uterus and brain of vaccinated guinea pigs showed no histopathological lesions (Figure 2). S19∆*per* vaccine candidate was proven safe for animal use.

### 3.2. Post Immunization Humoral Immune Response and DIVA Assessment

Prior to experimentation, seroreactivity of animals to *Brucella* antigen using RBPT was determined. None of the animals showed agglutination and were thereby considered free of *Brucella* infection. Post immunization seroreactivity was determined using RBPT and iELISA on serum samples collected at different time points. Spectra of IgG responses to immunization with S19, S19Δ*per* and S19Δ*per* reduced dose were recorded (Figure 3).

The S19 vaccinated animals showed significantly higher antibody titers in comparison to S19Δ*per* vaccinated animals (significant difference at *p* < 0.05). As early as 7 d post immunization, strong seroreactivity could be observed in all S19 vaccinated buffaloes by RBPT. Strong reactivity to RBPT antigen persisted for a long duration in the S19 group (Table 1). However, the majority of S19Δ*per* immunized buffaloes showed moderate, weak or no reactivity to RBPT antigen. Difference in seroreactivity patterns among vaccinated animals and clinically infected animals could be a DIVA indicator. DIVA status of S19Δ*per* was assessed after six weeks of immunization using RBPT. Serum samples collected from clinically infected animals showed strong agglutination on RBPT (data not shown). Similarly, all the animals immunized with S19 vaccine showed strong agglutination on RBPT; whereas, S19Δ*per* immunized animal sera showed lower degree of agglutination or no agglutination with RBPT antigen (Table 1).

### 3.3. Serum Biochemical Profiles of Vaccinated Animals

The serum biochemical profile was studied to determine any pronounced changes due to immunization. The relative quantities of the indicator enzymes and biomolecules were determined from the serum samples of the experimental animals from various groups. Normal levels of AST, ALT, AP, direct bilirubin, total bilirubin, CPK and creatinine were noticed and levels were non-significant between S19, S19Δ*per* normal dose (S19ΔP) and S19Δ*per* reduced dose (S19ΔPr) immunized groups (Table 2). However, mild hypoproteinemia and mild hypoalbuminemia was noticed in S19 and S19Δ*per* immunized groups.

### 3.4. In Vitro and In Vivo Persistency of Brucella Vaccine Strains

The effectiveness of in vitro complement lysis of the *Brucella* vaccinating strains was determined using sera collected from different groups of buffalo calves. Live *Brucella* S19 and S19Δ*per* organisms were exposed to fresh serum complement harvested from twelve seronegative buffalo calves. The number of viable organisms after exposure to serum complement was determined by colony count on *Brucella* agar plate. The S19Δ*per* was highly sensitive to buffalo serum complement in comparison to its parent strain S19 (Figure 4A). The S19Δ*per* was more efficiently lysed by buffalo serum complement (*p* < 0.0001). *B. abortus* S19 strain was more resistant to complement mediated killing and the average viable count was 213.0 ± 31.46, while the viable count for S19Δ*per* strain was 11.25 ± 4.573.

The persistency of vaccinating organisms in spleen, liver and lymph nodes were determined after 42 d of immunization. The bacterial load of S19Δ*per* strain was low as compared to the wild type strain S19 (Figure 4B). The liver was free from bacterial burden at least the time point when the enumeration was performed. Similarly, no bacteria were isolated from the spleen of vaccinated animals.

### 3.5. Gross and Histopathological Lesions

The gross lesions observed in spleen have been shown in Figure 5A–D. The spleens from S19, S19Δ*per* normal dose and S19Δ*per* reduced dose vaccinated groups were normal. However, one animal from the S19 vaccinated group showed congested serosal blood vessels and pinpoint hemorrhages (Figure 5B). However, tonsil, lungs, liver, kidneys, uterus, ovaries and brain from all vaccinated groups were normal and showed no pathological changes (data not shown).

Histopathological lesions observed in different organs are shown in Figure 6A–L. The S19 vaccinated buffaloes showed marked lymphoid hyperplasia with germinal center formation in spleens (Figure 6A). Parotid, prescapular, retropharyngeal, mandibular and supramammary LNs showed marked hyperplasia of lymphoid follicles with germinal center formation. Liver showed normal central vein with hepatic cords and normal sinusoidal space. However, one animal from S19 vaccinated buffalo showed multi-focal inflammatory cells infiltration in hepatic parenchyma and necrosis and apoptosis (chromatin condensation and apoptotic bodies) of granulosa cells, and reduced amount of follicular antral fluid in early tertiary follicles of the ovary (Figure 6G,J). Buffaloes vaccinated with S19Δ*per* showed marked lymphoid hyperplasia with germinal center formation in spleens (Figure 6B,C). Different lymph nodes including supramammary LN showed marked hyperplasia of lymphoid follicles with germinal center formation. The number of lymphoid follicles in spleens and LNs were almost equal in normal and reduced dose of S19Δ*per* vaccinated groups when compared to the S19 vaccinated group. Livers showed normal central veins with hepatic cords and sinusoidal space. A portion of the wall of a tertiary or Graafian follicle of ovary showed normal theca externa, interna and granulosa cells.

### 3.6. Assessment of Protective Efficacy of S19Δper Vaccine in Buffaloes

#### 3.6.1. Postmortem Examination

After the challenge with virulent strain 544, gross lesions were assessed by postmortem examination (Figure 7A–H). Unvaccinated control buffaloes upon virulent challenge showed enlarged spleens with marked thickening of the capsule especially at the borders, oedema, firm in texture and prominent serosal blood vessels due to congestion (Figure 7A). The parotid LNs were markedly enlarged and edematous with focal necrotic areas in the cortex and increased cortical area than medulla. On cut sections, watery exudate oozed out and bulged surfaces were noticed (Figure 7E). Markedly enlarged and edematous prescapular, retropharyngeal, mandibular and supramammary LNs with severe congestion of both cortical and medullary areas were noticed (data not shown). The S19 and S19Δ*per* vaccinated buffaloes showed moderately enlarged spleen and lymph nodes. However, one animal from S19Δ*per* reduced dose group showed mild thickening of the capsule especially at the borders and moderate congestion (Figure 7D). The liver, ovaries, uterus, kidneys and lungs from the S19 and S19Δ*per* vaccinated buffaloes were normal.

#### 3.6.2. Clearance of Challenge Infection

The protective efficacy of the *Brucella* vaccine can be determined by comparing the challenge bacterial load in the organs of the immunized animals against non-vaccinated animals. The non-vaccinated animals were only inoculated with PBS buffer and infected with a heavy load of challenge *Brucella* strain S544 (Figure 8). A significant reduction in S544 bacterial load was observed in the S19 and S19Δ*per* immunized animals (*p* < 0.05). Normal dose as well as reduced dose of immunization with S19Δ*per* conferred equivalent protection. Some of the LNs from S19-vaccinated buffaloes showed higher bacterial counts; however, the values were not significant. Interestingly, the liver was free of infection among all the vaccinated groups.

#### 3.6.3. Histopathological Lesions in Post Challenge Animals

Histopathology on tissues of post-challenged animals was performed (Figure 9A–P). Tissue sections of PBS control buffaloes showed splenitis with marked thickening of the capsule. Blood vessels in the capsule are prominent and dilated due to congestion. Lymphoid depletion with apoptosis and necrosis of the lymphocytes were noticed in white pulp (Figure 9A). Neutrophilic infiltrations were noticed in the subscapular and trabecular sinuses, and red pulp areas. Lymphadenitis and severe congestion of blood vessels in cortex and medulla of parotid LN were noticed (Figure 9E). Lymphoid depletion with apoptosis and necrosis of lymphocytes was observed in germinal centers of the lymphoid follicles in parotid and prescapular LNs. The uterus showed marked congestion of blood vessels and hemorrhages in the submucosal and muscular layers (Figure 9M).

Histopathology of S19 (Figure 9B) and S19Δ*per* (Figure 9C) immunized buffaloes showed lymphoid hyperplasia with germinal center formation in spleens. The number of lymphoid follicles in spleen and LNs were almost equal in normal and reduced doses of S19Δ*per* vaccinated groups when compared to S19 vaccinated group. The endometrium and endometrial glands were normal in all the vaccinated groups. Further, liver, ovaries, uterus, kidneys, lungs and brain from S19 and S19Δ*per* vaccinated animals showed no histopathological changes (data not shown). However, one animal from S19Δ*per* reduced dose group showed thickening of capsule due to oedema and moderate lymphoid depletion in spleen and LNs. The endometrium showed moderate congestion in the muscularis layer.

## 4. Discussion

Vaccination is the most effective way to reduce the disease burden and remains the central point to brucellosis control program. Treatment of *Brucella* infected animals with antibiotics is not recommended. Prolonged treatment regimen, requirement of higher concentration of antibiotics and relapse of infection limit the use of antibiotics in animals [18,19,20,21,22]. *Brucella abortus* S19 vaccine has been used effectively in different countries across the globe. However, major drawbacks of S19 vaccine in terms of its safety and interference in serodiagnosis led to numerous attempts to generate new strains, which are safer, elicit strong immunity and possess DIVA property [23,24,25].

In this study, we report the effective mitigation of *Brucella* infection in buffaloes by immunizing with live attenuated *B. abortus* S19∆*per* strain. A brucellosis control program has been initiated in India by the Department of Animal Husbandry Dairying and Fisheries during 12th five-year plan. Vaccination of the female calves at 6–8 months with live S19 plays a pivotal role in this program. Although safety is an issue, it is well recognized that single dose calfhood vaccination with live attenuated *B. abortus* S19 strain confers lifelong immunity in bovines [26]. Contrarily, killed whole cell vaccine and subcellular fractions are superior in terms of safety, but fail to elicit strong and long-term immunity [27,28,29,30,31].

Before conducting the vaccine trial in buffaloes, a safety assay for S19∆*per* was performed in guinea pigs. Vaccine seed culture must conform to the minimum set standards of the safety parameters. In this study, we followed the recommendation of the Indian Pharmacopoeia (IP). Any new vaccine candidate for bovine brucellosis must be compared with the reference S19 vaccine. Kinetics of humoral immune responses elicited after immunization was measured by iELISA. Interestingly, all the vaccinated groups showed seroconversion upon vaccination. Seroconversion being an indicator to assess successful vaccination could be suitably measured in S19∆*per* vaccinated animals.

Strong agglutination reaction was observed by RBPT assay. On the other hand, there were individual variations in RBPT reactivity in S19∆*per* vaccinated buffaloes. Most of the S19∆*per* vaccinated animals elicited moderate to weak agglutination. This was an expected outcome, because S19∆*per* possessed an altered LPS with residual O-polysaccharide (OPS) [14]. Reactivity to RBPT in S19∆*per* vaccinated group significantly decreased with the time. At 6 weeks post immunization, the majority of buffaloes vaccinated with S19∆*per* could convincingly be differentiated from clinically infected as well as from S19 vaccinated animals.

The DIVA feature would allow effective use of the test-and-segregation or test-and-slaughter policy. The DIVA diagnostic test or a DIVA-enabled *Brucella* vaccine is of prime importance in the brucellosis control and eradication program. *B. abortus* strain RB51 lacks O-side chain and possesses rough morphology. Vaccination with RB51 does not produce antibodies to the OPS antigen and is therefore undetected by conventional serological tests [32,33]. Further, RB51 vaccination in buffaloes showed less interference to diagnostic serology compared to clinical infection [34]. On DIVA concept, unique oligosaccharides prepared through chemical synthesis used as diagnostic reagent could also discriminate between brucellosis and infections caused by several bacteria with OPS that share some structural features with those of *Brucella* [35,36,37,38,39].

The complement system plays a dominant role in the innate immune system of the body and acts as the first line of defense against invading pathogens. However, invading bacterial pathogens have in turn evolved ingenious strategies to surmount complement activity [22]. The LPS mediates *Brucella* to colonize and results in a chronic infection in host cells [40]. Complement killing helps in clearing *Brucella* cells that are released into the blood stream. Smooth strains of *Brucella* contain OPS as part of their LPS structure, while rough strains lack the OPS [41,42,43,44,45]. The OPS also act as a virulence factor and inhibit complement-mediated cell lysis [41,46]. Therefore, smooth *Brucella* strains are generally more virulent than rough strains [41,43]. Increased sensitivity to complement is indicative of diminished residual virulence in microorganisms. The S19∆*per* has shown high sensitivity to buffalo serum complement. Therefore, S19∆*per* proves to be more attenuated and a safer *Brucella* vaccine candidate over the S19 vaccine. Although S19∆*per* has the added advantage of the safety parameters, to be considered as a potential vaccine candidate, it needs to commensurate with the inherent ability of conferring protection against infection. The LPS deficient mutants have been demonstrated to be highly attenuated in a number of studies, but in many instances, they failed to induce protective immune responses [47,48].

We conducted two immunization experiments in buffaloes and compared the results among the immunized groups. In the first experiment, we examined gross pathological changes in different organs and persistence of S19∆*per* in liver, spleen and different lymph nodes. The spleen and LNs are the primary target organs of the *Brucella* infection [49,50,51,52]. After 42 d of immunization, different organs from all the vaccinated groups were found normal and showed no gross pathological changes on necropsy (data not shown). Absence of lesions in predilected organs of *Brucella* infection indicated the safety of S19Δ*per* vaccine candidate for animal use. Both S19∆*per* and S19 were completely cleared from liver and spleen after 30 days challenge infection. However, there was significant difference in live bacterial count of S19 and S19∆*per* in LNs. Although there were variations among individual animals, the results were indicative of fast and efficient clearance of S19∆*per* infection from the lymphoid organs compared to S19 infection. This comparative study affirms the relative safety of S19∆*per* over to its parental strain S19. A major concern on the use of S19 vaccine is its persistence into adulthood [53,54]. Calves vaccinated with RB51 rough strain could rapidly clear the vaccine organisms from the prescapular lymph node [49] and these findings were similar to the observations of present study.

Protective efficacy of S19∆*per* was assessed by challenge infection with *B. abortus* S544 strain using previously published methods [55,56]. There were no significant gross lesions observed in different organs of vaccinated animals collected at necropsy. However, marked thickening of capsules in spleen and markedly enlarged, edematous, and focal necrotic lymph nodes were noticed in unvaccinated buffaloes after challenge infection.

In the present study, lymphoid depletion was noticed in LNs and spleen of unvaccinated and *B. abortus* 544 infected buffaloes due to apoptosis and necrosis of lymphocytes. It has been shown that *B. abortus* induced apoptosis of T lymphocytes [57]. Further, infection with a virulent strain of *B. abortus* S2308 resulted in severe histopathological lesions in lymphoid organs including lymphoid depletion [49]. In this study, we observed an efficient clearance of *B. abortus* 544 from the lymphatic systems, which might have resulted in less histopathological lesions in vaccinated animals. The hallmark of vaccine potency and efficacy is the ability to clear the challenge infection from various predilected tissues and organs. The challenge dose used was sufficient to set infection and to measure the spread of infection in buffaloes. However, challenge organism could not be isolated from one animal of control group. The reason is not known and this could be an error during the challenge infection or an individual variation wherein, this particular animal had resistance to *Brucella* infection. We observed generalized infection of S544 in unvaccinated animals. Our results suggest that S19∆*per* immunization confers protection from *Brucella* infection at a similar level to the reference S19 vaccine. It has been reported that calves vaccinated with 0.05 dose of S19, showed protection similar to normal dose but calves were not protected when 0.001 dose of vaccine was given [55]. The other reference vaccine, RB51, was proven safe and effective in cattle but the use of RB51 at the same dose in water buffaloes was found ineffective [58,59]. In another study, buffaloes were administered with a triple dose of RB51 vaccine and followed by a booster dose. The animals were challenged after one month of last immunization and did not induce infection [52].

Shedding of vaccinating organisms is a major concern when live attenuated vaccines are used [8,16,26]. There was no shedding of S19∆*per* in different body secretions examined after 7 and 15 d of immunization. However, we did not investigate the possibility of shedding during pregnancy or in the birth fluids after parturition. On a similar note, pregnant swine immunized with S19 and S19∆vjbR mutant resulted in seropositivity; however, shedding of organisms was not observed in the vaginal secretions [60].

## 5. Conclusions

*Brucella abortus* S19∆*per* efficiently colonized and elicited immune response in buffalo calves. The S19∆*per* vaccine candidate has been proven superior, wherein it was found safer than S19 strain and conferred protection against challenge infection. Unlike the S19 vaccine, the S19∆*per* immunized animal could possibly be distinguished from clinically infected animals by a commonly employed serological test, RBPT. We have further observed that a reduced S19∆*per* vaccine dose, at least 1/20^th^ of normal dose, could well be used in calfhood vaccination without compromising the level of protection. This would not only reduce the load of antigen in the vaccine formulation but also reduce the cost of production. Further, a reduced dose vaccine regimen would be safer and less stressful to the animals. However, vaccine efficacy and DIVA capability of S19∆*per* need to be investigated under field conditions involving large numbers of animals.

## Figures and Tables

**Figure 1 vaccines-09-01423-f001:**
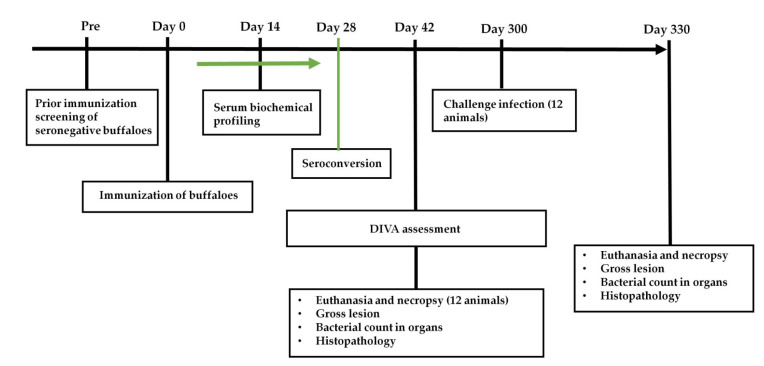
Immunization, sample collection and investigation scheme.

**Figure 2 vaccines-09-01423-f002:**
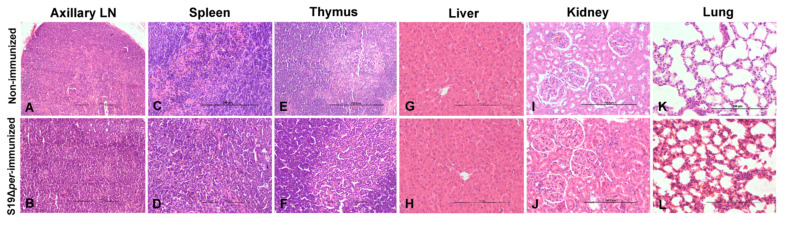
Histopathological lesions of vaccine safety assay in guinea pigs. (**A**) Axillary lymph node showed normal cortex and medullary areas. (**B**) Axillary lymph node showed lymphoid hyperplasia in cortex. (**C**) Spleen showed normal red and white pulp areas. (**D**) Spleen showed lymphoid hyperplasia in white pulp. (**E**,**F**) Thymus showed dark stained cortex and light stained medullary areas. (**G**,**H**) Liver showed normal architecture with hepatocytes arranged in radiating cords and normal sinusoidal space. (**I**,**J**) Transverse section of cortex of kidney showed normal glomeruli and proximal convoluted tubules. (**K**,**L**) Lungs showed normal alveoli lined by simple squamous epithelium. Tissue sections were stained with H&E staining and images were acquired at 100× (**A**,**B**,**E**) and 200× (**C**,**D**,**F**–**L**) magnifications using light microscope.

**Figure 3 vaccines-09-01423-f003:**
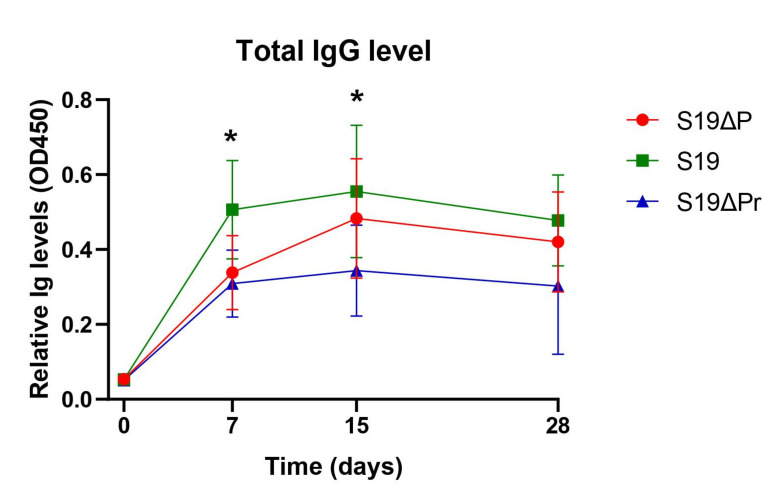
Total IgG levels among immunized buffaloes. Indirect ELISA based quantification of serum IgG concentration was performed. Statistically significant difference (* *p* < 0.05) was observed between S19 vaccine and candidate mutant vaccines, S19ΔP (S19Δ*per* normal dose) and S19ΔPr (S19Δ*per* reduced dose). Results are presented as a line diagram with mean ± SD at each time point. The difference in the titer of IgG was noticeable as early as on 7 days post immunization (dpi) and peaked at 15 dpi.

**Figure 4 vaccines-09-01423-f004:**
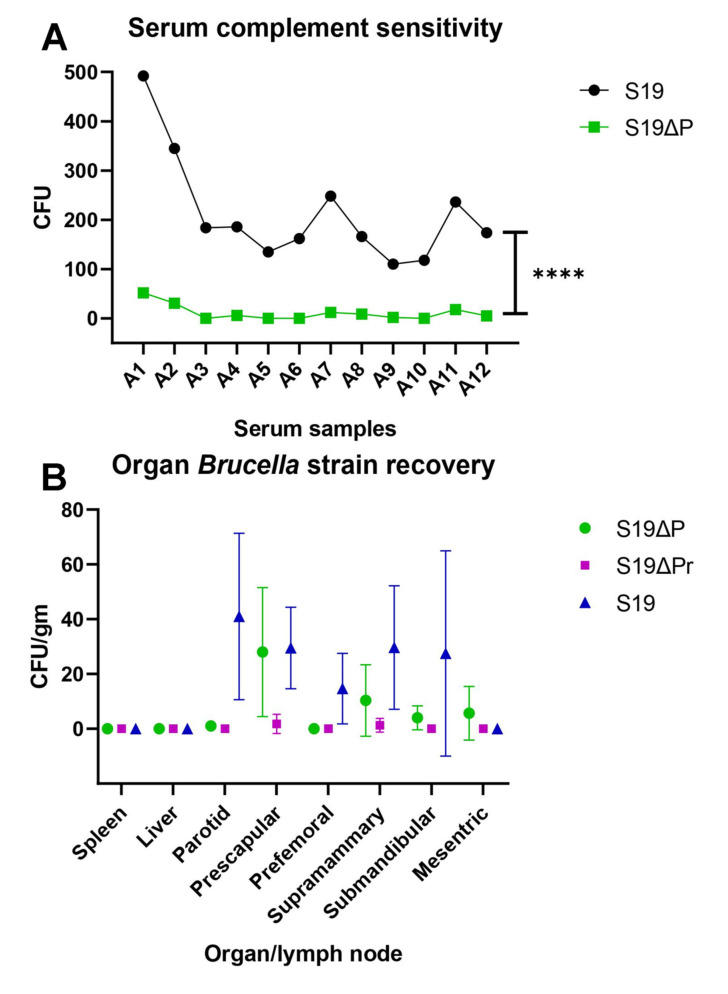
Persistency of *Brucella* vaccine strains in vitro and in vivo. (**A**) Serum complement sensitivity assay. The susceptibility of the *Brucella* strains against complement killing was determined. The S19 strain was relatively resistant to complement killing, while S19Δ*per* strain was highly susceptible to complement killing. The test was performed using fresh serum complement harvested from 12 buffalo calves (A1–A12). **** *p* ≤ 0.0001 by paired *t*-test. (**B**) *Brucella* vaccinating strain recovery from organ-wise. The residual virulence of S19Δ*per* candidate vaccine was measured by the bacterial burden in the vaccinated animals. At 42 dpi, load of S19Δ*per* was lower than S19 strain in inoculated animals.

**Figure 5 vaccines-09-01423-f005:**
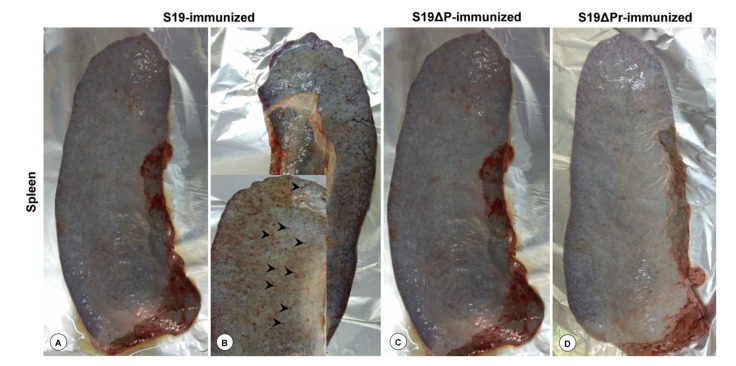
The gross lesions on the representative spleens of immunized buffaloes were shown. Spleens of animals immunized with S19 (**A**), S19Δ*per* (**C**) and reduced dose of S19Δ*per* (**D**) showed no observable gross pathological changes. One S19 immunized animal (**B**) showed congested serosal blood vessels and pin point hemorrhages (indicated by arrowheads).

**Figure 6 vaccines-09-01423-f006:**
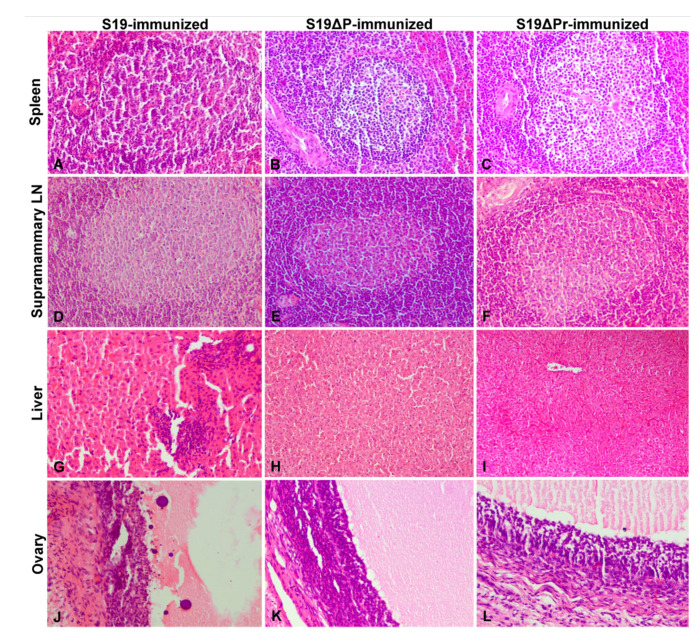
The histopathological lesions in various organs of immunized buffaloes are shown. (**A**–**C**) Spleen showed lymphoid hyperplasia with germinal center formation in white pulp. (**D**–**F**) Supramammary LN showed marked hyperplasia of lymphoid follicles with germinal center formation in cortex. (**G**) Liver showed multi-focal inflammatory cell infiltration. (**H**,**I**) Liver showed normal central vein with hepatic cords and normal sinusoidal space. (**J**) Ovary showed necrosis and apoptosis (chromatin condensation and apoptotic bodies) of granulosa cells, and reduced amount of follicular antral fluid in an early tertiary follicle. (**K**,**L**) Portion of the wall of tertiary or Graafian follicle showed normal theca externa, interna and granulosa cells of ovary. Tissue sections were stained with H&E staining and images were acquired at 200× magnification using light microscope.

**Figure 7 vaccines-09-01423-f007:**
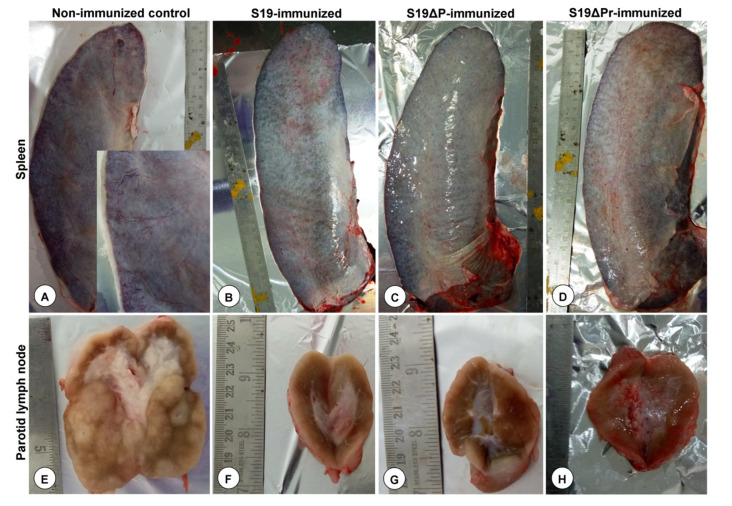
Post-challenge assessment of the gross lesions in the immunized buffaloes. (**A**) Enlarged spleen with marked thickening of capsule especially at the borders, oedema, and congested serosal blood vessels (inset). (**B**–**D**) Moderately enlarged spleen. (**E**) Markedly enlarged parotid lymph node (LN). Cut section revealed bulged surfaces with focal necrosis and increased cortical area than medulla. (**F**–**H**) Moderately enlarged parotid LN.

**Figure 8 vaccines-09-01423-f008:**
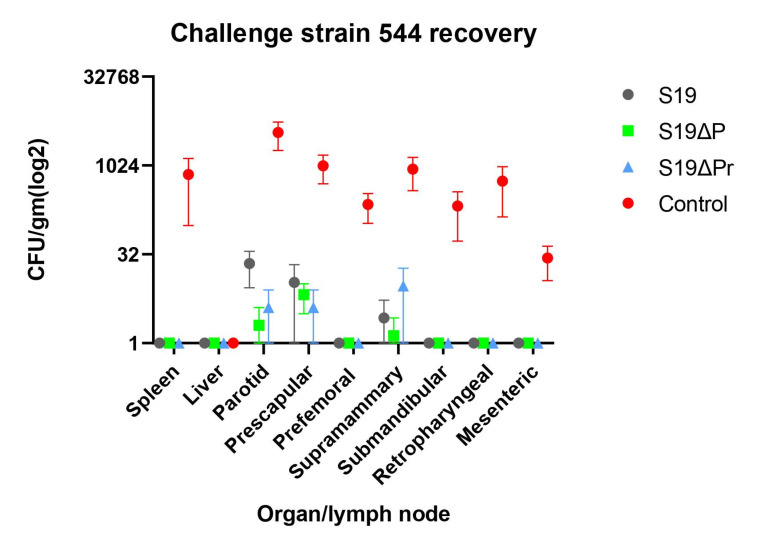
Post-challenge load of bacterial infection in organs and lymph nodes of buffaloes. Non-immunized control animals showed heavy Brucella burden in various organs, except livers. The animals immunized with candidate vaccine, S19Δ*per* and S19Δ*per* reduced dose, showed reduction in *Brucella* burden, which is comparable to standard S19 vaccine. Compared to non-immunized control group, a significant reduction in the *Brucella* count was observed in S19, S19Δ*per* and S19Δ*per* reduced dose groups. Within the immunized groups, the bacterial counts in different organs were not significantly different.

**Figure 9 vaccines-09-01423-f009:**
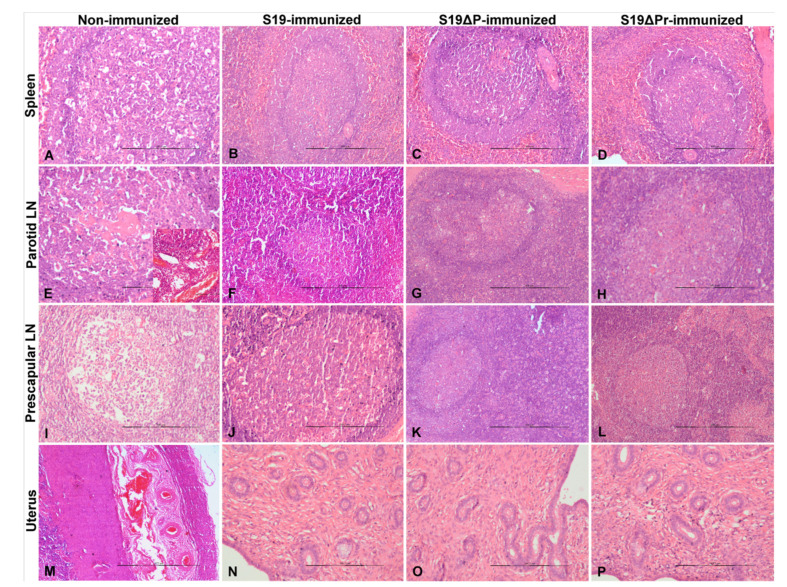
Post-challenge assessment of immunized buffaloes: histopathological lesions. (**A**) Spleen showed lymphoid depletion with apoptosis and necrosis of lymphocytes in white pulp. (**B**–**D**) Lymphoid hyperplasia with germinal center formation in white pulp. (**E**) Parotid lymph node (LN) showed lymphoid depletion with apoptosis and necrosis of lymphocytes in germinal center of lymphoid follicles. Severe congestion of blood vessels in cortex and medulla (inset). (**F–H**) Parotid LN showed hyperplasia of lymphoid follicles with germinal center formation in cortex. (**I**) Prescapular LN showed severe lymphoid depletion in germinal center of lymphoid follicles. (**J**–**L**) Prescapular LN showed hyperplasia of lymphoid follicles with germinal center formation in cortex. (**M**) Uterus showed marked congestion and dilatation of the blood vessels in muscular layer. (**N**–**P**) The normal endometrium is lined by ciliated single layer of columnar epithelium and endometrial glands. The tissue sections were stained with H&E staining and images were acquired at 40× (**M**), 100× (**B**,**C**,**F**,**G**,**I**,**K**,**L**) and 200× (**A**,**D**,**E**,**H**,**J**,**N**–**P**) magnification using light microscope.

**Table 1 vaccines-09-01423-t001:** Serum biochemical profiles in immunized buffaloes on 14 days post immunization.

	AST (U/L)	ALT (U/L)	AP (U/L)	Total Protein (g/dL)	Albumin (g/dL)	Direct Bilirubin (mg/dL)	Total Bilirubin (mg/dL)	CPK (U/L)	Creatinine (mg/dL)
S19	80.25 ± 4.10	27.00 ± 1.83	416.39 ± 48.38	5.46 ± 0.09	1.75 ± 0.15	0.31 ± 0.05	0.38 ± 0.03	143.46 ± 14.69	1.60 ± 0.17
S19Δ*P*	79.38 ± 4.10	29.00 ± 1.83	426.35 ± 48.38	5.22 ± 0.09	1.88 ± 0.15	0.45 ± 0.05	0.24 ± 0.03	134.62 ± 14.69	1.43 ± 0.17
S19ΔPr	81.25 ± 4.73	24.33 ± 0.88	405.33 ± 36.60	5.73 ± 0.53	1.87 ± 0.15	0.33 ± 0.12	0.43 ± 0.05	153.22 ± 7.58	1.34 ± 0.23

The values from different groups did not show any significant differences. AST—aspartate amino transferase; ALT—alanine amino transferase; AP—alkaline phosphatase; CPK—creatine phosphokinase; S19ΔP—S19Δ*per* normal dose; S19ΔPr—S19Δ*per* reduced dose.

**Table 2 vaccines-09-01423-t002:** Seroreactivity and DIVA status of S19, S19Δ*per* and S19Δ*per* reduced dose immunized animals analyzed by rose bengal plate test (RBPT).

Vaccine Group	Animal No.	Days Post Immunization (dpi)							
		0	7	14	28	42	70	112	300
S19	1	−	+++	+++	+++	+++ *	X	X	X
	2	−	++	+++	+++	++ *	X	X	X
	3	−	+++	+++	+++	+++ *	X	X	X
	4	−	++	+++	+++	+++ *	X	X	X
	5	−	+++	+++	+++	+++	++	+	−
	6	−	+++	+++	+++	+++	++	+	−
	7	−	+++	+++	++	++	+	−	−
S19Δ*per*	1	−	−	+	−	− *	X	X	X
	2	−	++	++	++	++ *	X	X	X
	3	−	++	++	+	+ *	X	X	X
	4	−	+	+	+	− *	X	X	X
	5	−	+	++	+	+	−	−	−
	6	−	+	+	+	+	−	−	−
	7	−	++	++	+++	+++	++	−	−
S19Δ*per* reduced dose	1	−	−	−	−	− *	X	X	X
	2	−	−	−	−	− *	X	X	X
	3	−	−	+	+	− *	X	X	X
	4	−	+	+	+	+ *	X	X	X
	5	−	−	−	−	−	−	−	−
	6	−	+	+++	+++	++	−	−	−
	7	−	++	++	++	++	−	−	−
PBS control	1	−	ND	ND	−	ND	ND	ND	−
	2	−	ND	ND	−	ND	ND	ND	−
	3	−	ND	ND	−	ND	ND	ND	−
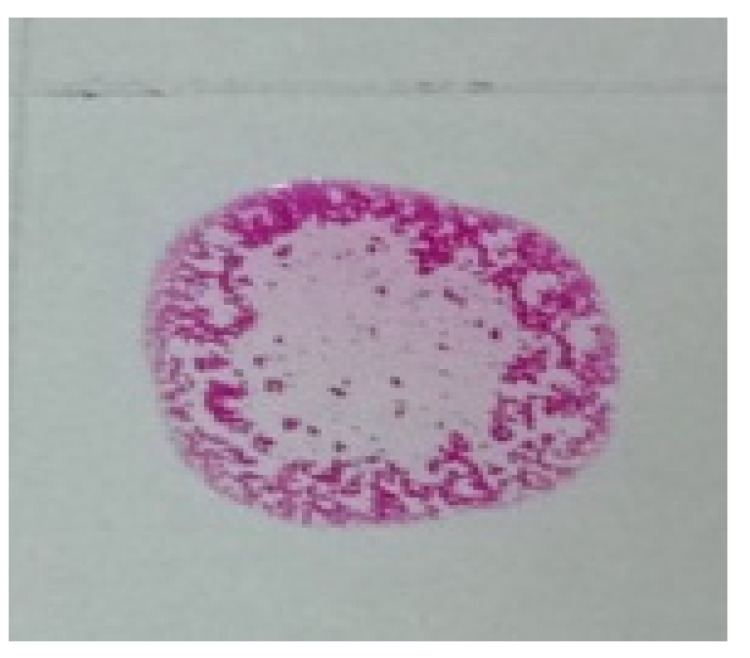	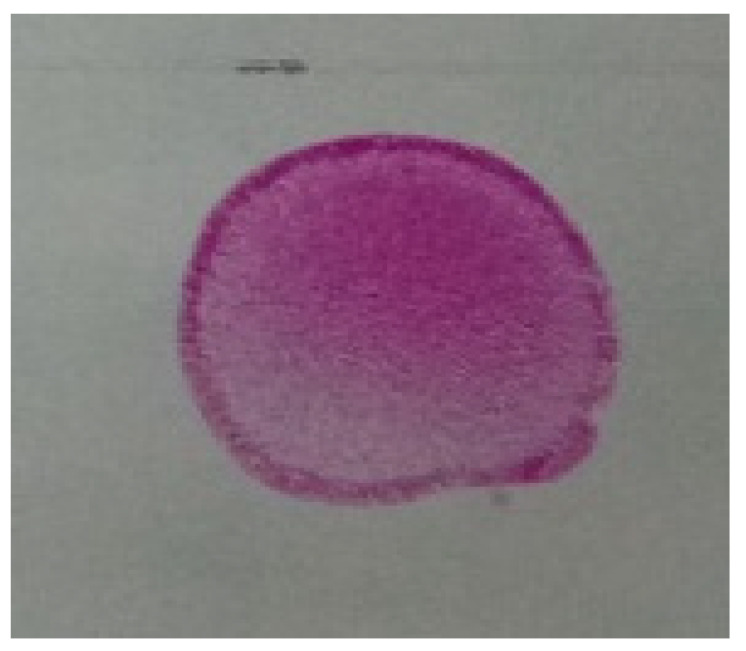			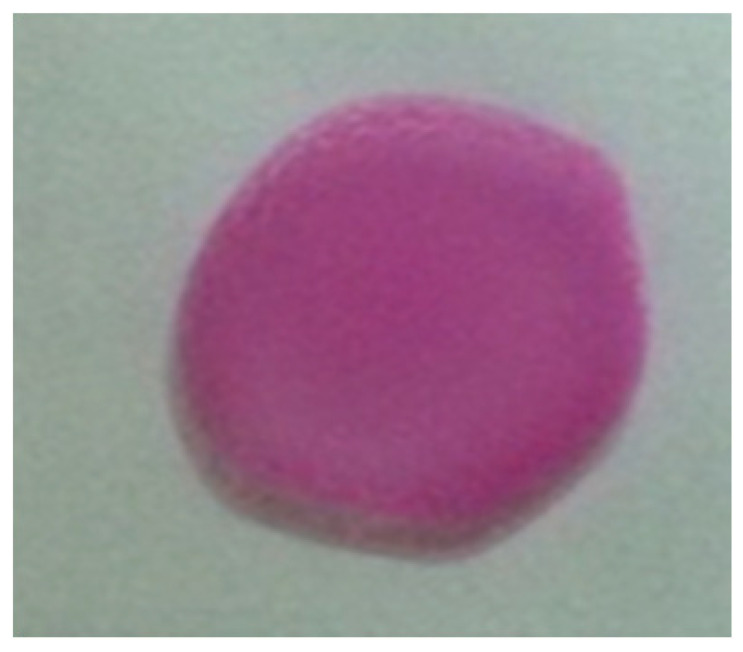			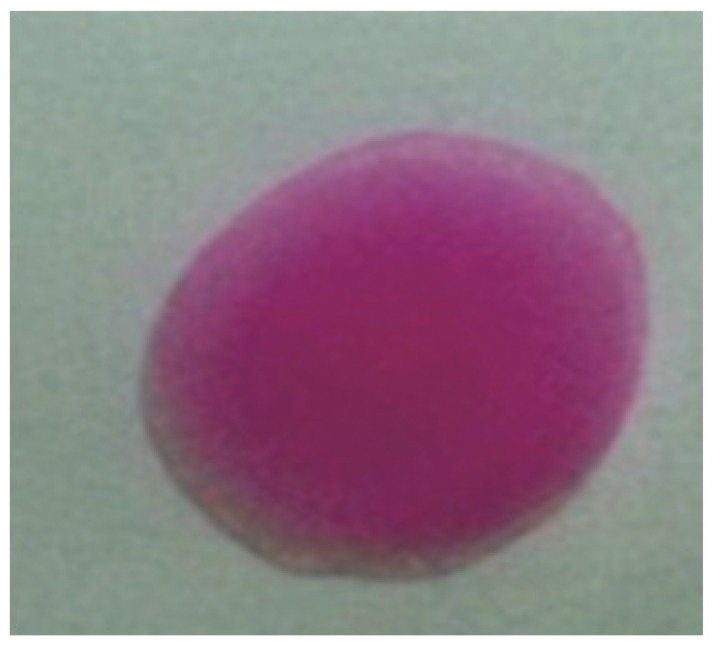		
+++	++			+			−		

Representative figures of RBPT reaction. RBPT strong reactivity: +++; moderate reactivity: ++; weak reactivity: +; no reactivity: −; animal sacrificed on 42 days post immunization (dpi): *; ND: not done; X: animals sacrificed on 42 dpi and serum not available.

## Data Availability

The data presented in this study are available within the article.

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
