# Peer review of "Immunization with Brucella abortus S19Δper Conferred Protection in Water Buffaloes against Virulent Challenge with B. abortus Strain S544"

_vaccines, 2021, doi:10.3390/vaccines9121423_

Round 1
Reviewer 1 Report
This study by Chaudhuri et al titled “ Immunization with Brucella abortus S19Δper Conferred Protection in Water Buffaloes against Virulent Challenge with B. abortus Strain S544” studied the potential of Brucella abortus S19Δper vaccine as vaccination with S19 has some draw backs. This is a well done study with appropriate research design. I have couple of concerns regarding this article
- As the number of cattle used in each group were small, how many times this study was repeated?
- Can authors please edit discussion part, as lot of it looks similar to introduction?
Apart from this I would like to appreciate Chaudhuri et al for their work.
Author Response
- We agree with the reviewer on that the number of animals used in the trial is lower. In this study, we have used a total of 24 buffalo calves. The study needs sacrifice of animals at the completion of experiments. There is a very stringent regulation to use large animals for experiment in India. For a pilot-large animal trial, our observations revealed that the number of animals that was used is sufficient to differentiate between the control and vaccinated groups. The present study is a follow up to our pre-clinical trial where we have employed larger number of small-experimental animals (Ref. 14). Our previous study on mice model has proven the efficacy of S19∆per as vaccine candidate (Reference no. 14) and thereby we could get approval to conduct experiment on a minimum required number of buffalo calves. The present experiments have been designed to prove the immune responses, safety and response to challenge infection in natural host, buffaloes. The results clearly indicated the efficacy of vaccine candidate in target host. The results obtained in this study would be the baseline for taking up a vaccine field trial in large number of animals, reared under natural field conditions.
- According to the suggestions of the reviewer, the discussion part has been edited in the revised manuscript.
Reviewer 2 Report
This is a well-executed and nicely presented study with practical significance to animal agriculture in developing countries.
Please proofread the manuscript for grammatical correctness. I have added most of my comments and suggestions in the original PDF (attached).
Please address the questions accordingly.

Author Response
Abstract:
Line no. 26-27 – As suggested, corrections were made in the revised manuscript.
Line 28: Brucella vaccine normally contains 4 X 1010 cfu per animal given by subcutaneous route. We have also investigated the vaccine performance at lower dose, 1/20th of normal dose.
Introduction:
Line no. 49, 60, 61, 86 - As suggested, corrections were made in the revised manuscript.
Materials and Methods:
Line no. 90, 109-112, 146, 195, 200, 204, 208, 216 - As suggested, corrections were made in the revised manuscript.
Line no. 111 - Animals were observed 10 days post-challenge for development of any clinical signs and mortality. Included in the revised manuscript.
Line no. 121 – Same query answered at line no. 28
Line no. 127 – Company and place were included.
Line no. 140 – Culled word replaced with sacrificed as suggested.
Line no. 170-171 – As suggested, followed a uniform format (company, location, country).
Line no. 220-222 – As suggested, included more detains in the revised manuscript
Results:
Line no. 226 – No local adverse reaction observed at the site of injection. None of the animals showed any clinical sings or mortality. At the site of injection, sometimes mild swelling is observed, which subsides within 24 hours and considered as insignificant in terms of adverse reactions.
Line no. 252, 265, 266, 267, 289, 291, 292, 302, 305, 312, 326, 337-342, 349, 350, 359-362, 366, 375, 377, 378, 381-383, 387, 392 – As suggested, corrections were made in the revised manuscript.
Table 1. Line no. 263 – The values from different groups did not show any significant difference. This has been mentioned in the revised manuscript.
Figure 2. As suggested, corrections were made in the figure legend. Old figure is replaced with new one showing hyperplesia in lymph nodes and spleen; glomeruli and cortical tubules in kidneys.
Figure 3. Suggestions incorporated in the figure and corrections also made in figure legend.
Figure 4. Line no. 298 – Yes, recovery of Brucella vaccinating organisms from different organs. Corrections made in figure legend as suggested.
Figure 5. Line no. 309 – As suggested, a meaningful title provided and corrections made.
Figure 6. Line no. 329 – A better name for the figure title provided. Corrections made as suggested.
Figure 6. Line no. 332 – In figure 6J, yes it is apoptosis. The histopathology image had necrosis and apoptosis (chromatin condensation and apoptotic bodies) of granulosa cells, and reduced amount of follicular antral fluid in early tertiary follicles of ovary. Further, Brucella organisms and pathological lesions were reported in ovary of cattle in previous reports.
Figure 7. - Corrections made as suggested
Figure 8. - Corrections made as suggested
Figure 9. – Corrections made as suggested
Discussion:
All the suggested corrections have been incorporated in the revised manuscript. Another reviewer (Reviewer no. 3) suggested eliminating redundancy in the discussion. Therefore, a major editing has been made in the discussion in the revised manuscript.
Line no. 422 – Time information included in the revised manuscript.
Line no. 429-430 – Safety clearance of vaccine seed culture or vaccine lot is essential requirement for commercial use of vaccine in animals. Our data on safety will support the utility of S19∆per as a potential vaccine candidate.
Line no. 459 – Corrections made in the manuscript
Line no. 464 – The statements have been changed in the edited discussion, as suggested by the reviewers.
Line no. 469-472 – Reference added in the revised manuscript
Line no. 487 – The term lymph node first written in the manuscript as ‘lymph node (LN)’ and then subsequently used as an abbreviation ‘LN’ in the rest of the document
Line no. 487 – Reference added
Line no. 493 – After 30 days challenge infection. Suggested point has been mentioned in the revised manuscript.
Line no. 499-500 – The sentences were modified in a more meaningful manner.
Line no. 510-513 – Edited the discussion.
Line no. 539 – Reference added
Line no. 539 – Time information has been included.
Conclusions:
As suggested, editing has been done in the conclusion to specifically focus on the superiority of S19∆per as vaccine candidate.
Line no. 548 – Brucella abortus S19∆per is an intermediate phenotype strain because it is not completely rough or completely smooth. Instead, it shows apparently smooth colony forming morphology.
Reviewer 3 Report
The authors describe the Immunization with Brucella abortus S19Δper that might confer protection in Water Buffaloes against virulent challenge with B. abortus Strain S544. The manuscript is interesting while there are few concerns that need to be addressed:
- The manuscript has much grammar errors and need extensive English editing by native language speaker.
- In the methodology, please provide more details about the S19Δper; you can refer your previous study.
- For the safety studies, the authors used guinea pig model with 5X 109 cfu S19Δper, while for the vaccination formulation, they used 4X 1010 cfu, why?
- For the animal groups, I would rather recommend to draw a graph/figure to be easy for the readers and avoid confusion.
- For the vaccine formulation, what was the composition of this vaccine? Did the authors used any adjuvant and why if they didn’t?
- The authors mentioned that Serum samples were collected at specified time intervals; can you please clearly mention which time?
- From day 42 till 300, why the authors did not collect sera samples for seroconversion to monitor how was the progress of antibodies production?
- It is confused for the experimental layout between the challenge and persistency experiments please separate them into two different experimental layout plan not within the same figure.
- For the Brucella strain recovery, I noticed there is non-significant difference between the normal and low dose, what the authors think about that and why?
- There are too much redundancy between the introduction and discussion sections, please revise and edit.
- Why the authors did not screen any of the mucosal immune response (ISGs) by real time PCR for example?
- It is very important for the veterinarian in the field to know that one dose is enough to protect against Brucellosis and how long the immune response can elapse? In addition, did the authors think two dose will be much better and how long time in between?
- Conclusions, poorly written and need more editing.
Author Response
- Grammar errors have been corrected in the revised manuscript. Another reviewer (Reviewer no.2) has also suggested and indicated suitable grammar corrections in the manuscript. All corrections have been included in the revised manuscript.
- At 2.2 (Materials and Methods), as per suggestion of the reviewer, more detail on S19∆per has been provided in the methodology.
- Recommended dose for Brucella abortus S19 calfhood vaccination is 4 X 1010 cfu per animal. Whereas, safety of Brucella vaccine is tested in guinea pigs as per Indian Pharmacopoeia (IP). In accordance to IP guidelines, guinea pigs are injected with vaccine seed culture containing 5 × 109 viable organisms or 1/10th of calf dose given intramuscularly and observations are made for 10 days. Vaccine seed culture passes safety test when none of the animal show notable adverse reactions or death attributable to vaccination.
- Buffaloes were immunized and evaluated for in-vivo safety assay (persistency in organs, gross lesions and histopathological changes). Twelve animals were euthanized 42 days after immunization. This has been clearly represented in fig. 1. Other 12 animals were given challenge infection on day 300 after immunization. These animals were sacrificed 30 days after challenge infection (Fig. 1).
- Conventional Brucella abortusS19 vaccine is a live vaccine. Vaccine formulation is usually made in buffered saline solution with specific live count. No adjuvant is used in live Brucella vaccine. Similarly, vaccine formulation used in this study contained live abortus S19∆per strain and no adjuvant was used. Unlike subunit or killed vaccine, most of the live vaccines do not require adjuvant to elicit immune responses. Live vaccine confers stronger and long duration immunity.
- Serum sample collection time points have been mentioned in the revised manuscript.
- Seroconversion after vaccination could be observed as early as 7 days post-immunization (Fig. 3). This indicates the potential of vaccine candidate to mount a detectable quick immune response to vaccinated animals. Serum samples also collected between 42-300 days post immunization from 12 animals. Other 12 animals were sacrificed on day 42, post-immunization, so serum samples are not available for these animals. Seroreactivity was monitored by RBPT assay (Table 2). RBPT is the most employed/recommended test for seroreactivity and serosurveillance of brucellosis in animals in the field.
- At 2.4 (Materials and Methods) the two different experimental layouts, bacterial persistence assay and challenge infection, were clearly mentioned. However, two experiments were separately described under separate paragraphs. This has also been highlighted in the schematic representation (Fig. 1). Similar immunological studies were conducted on samples and therefore these two experiments were included in the same heading at 2.4. In the results (3.4 and 3.6 of Results section), we have separately depicted the findings.
- We agree with reviewer’s comment. The main goal of including a lower dosing in the present study was to see if there are any benefits of reducing the dosage while maintaining the useful properties of the vaccine. We expected that a reduced dose of immunization will be effective enough to resist Brucellachallenge study. This is not a very new observation. Low dose vaccination in adult animals is also recommended in certain situations. Usually live vaccine provides good immunity even in low dose. Live organisms able to colonize and replicates inside the body and thereby mount very strong immune responses. We have observed that DIVA capability with reduce dosage is promising (Table 2). From the commercial point of view, a low dose vaccine formulation will be more cost effective. This is specifically very important for the farmers and animal owners who mainly come from economically weak backgrounds.
- As suggested by the reviewer, the discussion section has been edited to reduce the redundancy with introduction.
- We have not used any experiment to analyze mucosal immunity in this study. However, serum based immunological studies and challenge infection clearly indicated the potential S19∆peras a safer and effective vaccine candidate for brucellosis.
- A single dose of live Brucellavaccine is practiced in India and most of the countries around the world. Live Brucella vaccine provides a solid immunity which last for very long duration. These are the merits of live vaccine. Adults are not usually vaccinated with live Brucella vaccine due to safety issue and interference with serodiagnosis of clinical infection. These are the major drawbacks of Brucella vaccine which have been addressed in this study. Field veterinarians are aware of this vaccine policy. A booster dose is not required for live Brucella Based on these findings, we would be permitted to go for limited field trial involving large number of animals.
- As per the suggestion of the reviewer, conclusion has been edited in the revised manuscript.
Round 2
Reviewer 3 Report
Thanks for considering my comments.